# An Optimized Planning Tool for Microwave Terrestrial and Satellite Link Design

Eduardo Ferreira [1,*], Pedro Sebastião [1,2], Francisco Cercas [1,2], Carlos Sá Costa [1,2] and Américo Correia [1,2]

1   Department of Political Science and Public Policy, ISCTE—Instituto Universitário de Lisboa, 1649-026 Lisbon, Portugal
2   Instituto de Telecomunicações, 1049-001 Lisbon, Portugal
*   Correspondence: efcmf@iscte-iul.pt

**Abstract:** Today, the internet is fundamental to social inclusion. There are many people that live in remote areas, and the only way to supply internet services is through the use of microwave terrestrial and satellite systems. Thus, it is important to have efficient tools to design and optimize these systems. In this paper, a tool with the objective to shorten the time spent in the design process of microwave terrestrial and satellite point-to-point links is presented. This tool can be applied in academia by engineering students, providing an extended analysis of many sections of a link project design, as well as in professional practice by telecommunication engineering departments, presenting a concise step-by-step interactive design process. This tool uses three-dimensional world visualization, with the Cesium Application Programming Interface (API), to display and analyze site-specific characteristics that can disrupt the link's quality of service (QoS). Using this visualization, two ray-tracing algorithms were developed to analyze signal diffraction and reflection mainly throughout terrestrial links. Using this new algorithm, an innovative process for signal diffraction and reflection calculations was created. Using updated standards provided by the International Telecommunication Union Radiocommunication Sector (ITU-R), the characteristics of the defined simulated links could be predicted, thus providing the user with the metrics of signal quality and system link budget.

**Keywords:** radio propagation; software planning tool; terrestrial microwave link; satellite microwave link





## 1. Introduction

Terrestrial and satellite radio frequency and microwave systems are commonly used to connect communication services, by providing a peer-to-peer link, and are used to transmit data from one point to another. They introduce advantages for long-distance communication, such as high portability, easy installation, and lower installation and operational costs. Currently, these systems are used to provide access to the internet and other communication services in rural and precarious regions [1–3], in addition to being used for everyday human activities and having multiple industrial applications [4].

The design of these types of systems is a methodical and time-consuming process that requires calculations such as attenuation, fading, margins, frequency planning, and evaluation of the line-of-sight (LoS) in a given link, following the International Telecommunication Union Radiocommunication Sector (ITU-R) standards. Given the increased number of variables involved, a perfectly precise solution becomes almost impossible to achieve, relying on module simulations [5]. To address the gaps between system specification, system simulation, and circuit-level simulation, modeling and simulation must accompany the design phases from specification to the overall system verification [6].

To analyze "what if" scenarios, a main estimation is made with calculations, allowing an evaluation of the reliability of the designed link. This estimate is defined as the link budget. Since the change in a single characteristic can impose itself on the whole link

budget, a recalculation of the ITU-R specification models is in order. As a consequence, a constant manual recalculation of this model becomes impractical for engineering fields.

In the academic field, the ability to provide an application of the methods studied can be a valuable asset in gaining a greater grasp, not only of the methods themselves but also of how they evolve as a result of a change in scenario. For this reason, being able to provide an easy-to-understand demonstration of the process required for the design of systems holds high educational value for engineering academic subjects.

Although tools to provide this service have already been developed and distributed, the available tools have several drawbacks such as unaffordable pricing for academic use (licenses can cost over USD 300 monthly [7]), outdated specifications in terms of standards used, and poor user experience.

Subsequently, a need arises for an all-in-one tool that can provide the principal features needed to assert system link budgets, available at an affordable price for students and without compromising on the educational value of intermediate computations. This leads to a reduction in the time it takes engineers to compute all the components of the link budget of a designed link by providing an easy-to-understand graphical user interface to be used by users with little to no instructions.

To assess the QoS of a communication link, the general steps presented in Figure 1 were established to develop the tool.

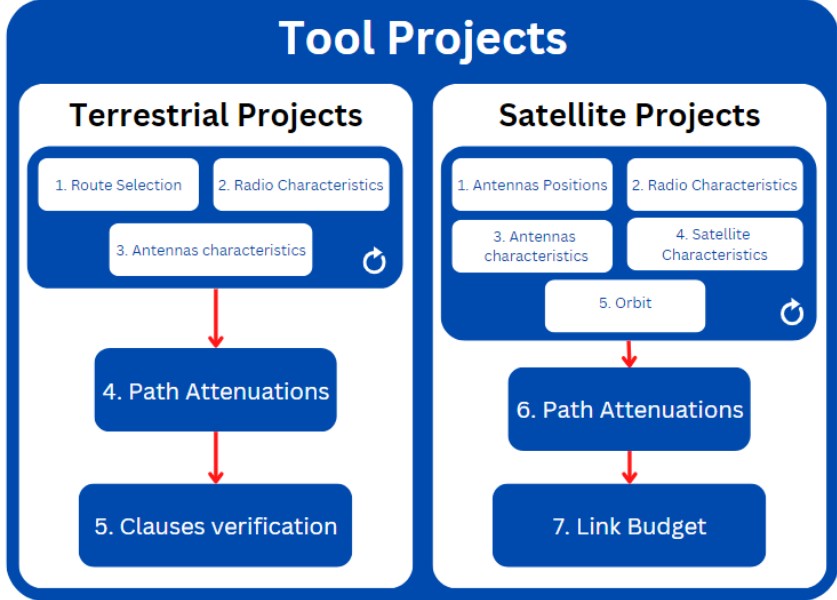

**Figure 1.** Developed Tool workflow.

As shown, the user must define the variables where the link will be implemented. This group of steps provides a vast number of iterations to find the optimal characteristics for the best QoS possible. For terrestrial communications, the route selection, antenna characteristics, and radio characteristics exert a major effect on the attenuation found through the link path, which will subsequently impact the metrics used to evaluate the link. In satellite communications, although most of the attenuations found in terrestrial systems are not applied, the choices of orbit and antenna positions (which will also determine the link connection distance) reflect major changes in the attenuations found and, as a consequence, in the necessary characteristics found in the link budget to make the designed link possible.

In order to present a tool including the most important features for radio-link planning, we performed an extensive search of the literature for currently available tools, comparing their benefits and drawbacks.

Out of these, we selected the following ones since they also have the same purpose as ours, both for academic and professional use:

- LINKPlanner, available in [8];
- TAP 7, available in [9];
- MLinkPlanner 2.0, available in [10];
- Feixer, presented in [11];
- Smart Link Planning Tool (SLPT), presented in [12,13].

The tools analyzed were developed both through academic investigation, for the case of Feixer and SLPT, and enterprise development, with allocated teams of developers to create the said tools, for the case of LINKPlanner, TAP 7, and MLinkPlanner 2.0. For paid licensing tools, the available free versions were used. The tools found in this case were TAP 7 (with a yearly license of USD 1999 [14]) and MLinkPlanner 2.0 (with a yearly license of USD 399 [15]).

Analyzing the presented tools, the characteristics denoted while using said tools are presented in Table 1.

**Table 1.** Comparison of the researched tools and the developed tool in this study.

| | User-Friendliness | Automatic Obstacle Detection | 3D Path Analysis | Urban Building Integration | Updated Standards | Single Software |
|---|---|---|---|---|---|---|
| LINKPlanner | | | | | x | x |
| TAP 7 | x | x | x | | x | x |
| MLink-Planner 2.0 | | x | | x | | x |
| Feixer | | x | | | | |
| SLPT | x | x | | | | |
| Proposed Tool | x | x | x | x | x | x |

## 2. Technology Used

A Web development technique was used to create a flexible and user-friendly software solution that also provides easier access to and interaction with third-party software. The suggested utility was created in this way utilizing the HTML, JavaScript, and CSS computer languages. The use of JavaScript, which is used to specify the processes that occur after user interactions, adds functionality to the visual structures that can be presented and interacted with by users using HTML. Finally, CSS was used for styling and animation, which helps the user understand the presented page.

The Electron Framework [16] was used to ensure a cross-platform, client-side application without the direct use of a Web browser. A developed Web page is shown with this framework using the Chromium browser in a way similar to that of most other web browsers, and the developed software is hosted on a virtual local server using Node.js. With the use of a package manager such as Node Package Manager (NPM), Node.js also enables the integration of the modules created by third parties (referred to as node modules). The Git and GitHub platforms were used for software version management. These platforms offer the management and storage of several produced feature versions, which is beneficial for software development [17].

## 3. Tool Description

### 3.1. Tool Structure

The software and application designing tool Adobe Xd was used to achieve the specified objectives. This software has drag-and-drop capabilities that make it easier to see the design that will be used. In order to achieve the aforementioned goals, a prototype design, shown in Figure 2, was made.

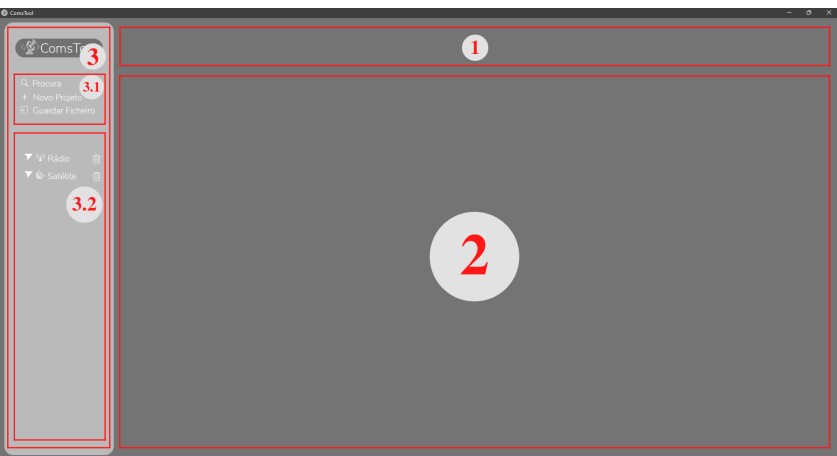

**Figure 2.** The developed tool's main structure.

The areas highlighted in Figure 2 represent the main objectives previously presented, where area 1 provides the selected project title (defined by the user on the project creation), area 2 presents the selected project content, area 3 contains a side menu with an easy process for the creation and saving of the projects and project progress in area 3.1, and the presentation of the created projects, allowing the user to select or delete a project in area 3.2.

*3.2. Project Creation*

To create a new project, after the user selects the necessary button to initiate this process, a pop-up box is presented, allowing the user to select the different types of projects available, the level of data presented, and the type of visualization to be presented. The level of data to be presented and the type of visualization are the two features resulting from client meetings.

Upon creating a project, a selector element is added to the side menu, where the user can select the project to load the created project. This element includes the project name, an icon to distinguish which type of project it contains, an icon allowing the user to delete the project, and an icon that, when pressed, provides a window on the side menu presenting all the steps of the said project and an icon that warns the user when the file contains elements that are not saved, being hidden when all the defined project elements are saved.

To simplify the process for users to interact when using a different type of project, the content of the various projects all adhere to the same design principles. With this layout, a series of windows that analyze various project creation and analysis processes are produced. This process is consistent with how a real communication project would typically proceed.

*3.3. File System Architecture*

A file system was developed to enable the user to save the current project progress for future work and to access it again. When the software is first run, a file folder is automatically established to achieve this purpose. This folder is generated in the operating system's application data directory, which varies depending on the operating system (for example, C:/Users/User Name/AppData/Roaming for Windows and./Library/Application Support for Mac-OS). Therefore, the Electron app API was used to enable the development of cross-platform apps by accessing this directory independently of the operating system.

The JSON file format was used to store the project's progress given the compatibility of JavaScript and JSON files. Upon loading the project page, the saved JSON object is saved in memory for easier access.

In the section in which the user chooses the project, an icon is displayed to show the user that a file has not been saved. When the project is saved, this icon is hidden to signify that the project's present state is saved. Finally, the project-specific file is only changed when the user chooses to save it. Given these characteristics, the file system architecture resembles

that of a non-relational database, where each project simply saves the user-defined variables that differ across different types of projects (terrestrial or satellite projects).

### 3.4. Terrestrial Links

In this section, the terrestrial-link design steps in the developed tool are presented.

### 3.4.1. Route Selection

To define the communication path, the Cesium API was used. The API allows the integration of external three-dimensional modules imported using a GL Transmission Format Binary (GLB) file format. The GLB format was used to exhibit the 3D models, as it represents a binary structure of the data included in GL Transmission Format (glTF) files, using JavaScript Object Notation (JSON). The Blender software, which enables the development of 3D models and the export of these models using the GLB file format, was used to build the GLB file for the antennas.

In order to track user clicks on the Cesium window, Cesium offers an internal window listener. Using the camera position and projection, it is possible to store and interpret the user's click location for a true place in the scene. After clicking the window, the user can distinguish between the components of the communication system by label and a 3D representation of an antenna placed on the spot they had chosen.

The path length and other measurements needed for calculations were taken straight from the Cesium API, as Cesium offers a 1:1 measurement of length in comparison to the real world.

If both the transmission and receiving antennas are in place, the first Fresnel Ellipsoid is represented between both antennas, as depicted in Figure 3. This ellipsoid provides an estimate of the path through which the signal energy travels. For the ellipsoid to be presented correctly, it is placed in the middle point between the path origin and the target and rotated using the Eye-Gaze-Up representation [18].

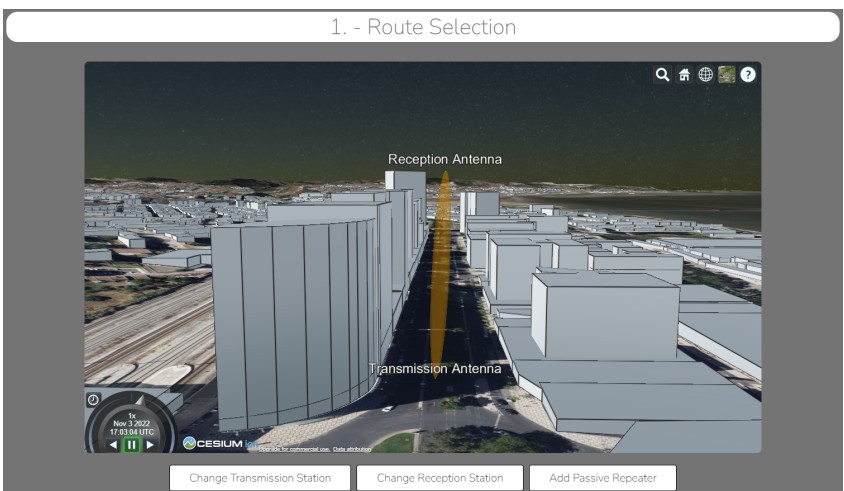

**Figure 3.** Path definition window.

Similar to how a station antenna is integrated into a communication system, so is a passive repeater. A passive repeater model is positioned once the user selects the inclusion of a passive repeater and specifies the location where it is to be placed. The orientation where the passive repeater is placed reflects the angle where the post-reflection signal can be found in the receiving antenna.

### 3.4.2. Antenna Characteristics

The user can change antenna characteristics such as pool height, dish diameter, and radiation efficiency. Using the data made available by Telewave [19], the user is presented with the option to select between different radiation patterns, as presented in Figure 4.

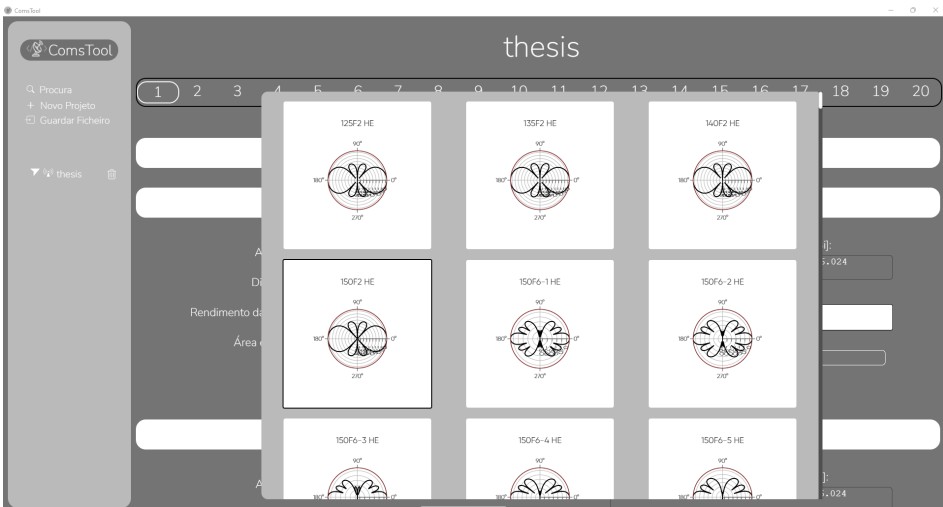

**Figure 4.** Radiation pattern window.

### 3.4.3. Radio Characteristics

In this design step, the user inputs all the radio parameters such as transfer rate, bandwidth, modulation, and frequency. The user can analyze multiple frequencies given a minimum and maximum frequency. The user is able to choose between quadrature amplitude modulation (QAM) and phase-shift keying (PSK) with the number of symbols inputted by the user. The user may also choose the polarization of the signal, having a horizontal and vertical option.

### 3.4.4. Signal Attenuation

The signal found in the receiving antenna is altered by the path in which it travels. Typically, these alterations result in signal attenuation that can be divided into the following elements:

- Free space attenuation;
- Diffraction attenuation;
- Atmospheric attenuation;
- Rain attenuation;
- Signal fading;
- Signal reflection.

Each step represents a different window on the project page, where the user may insert necessary information (such as polarization, for example) or change automatically acquired information (such as rain intensity). The global real-life elements necessary for signal attenuation calculations are acquired by two different processes. Either by the use of the OpenWeatherMap API [20], used for real-time weather variable acquisition, or using the digital maps provided in multiple ITU-R Recommendations. These digital maps may vary in precision and meaning; whereas one digital map has leaps between the latitude and longitude angles of 0.75 degrees, other digital maps may have an angle leap of 1 degree. For a faster search of the values in these digital maps, the content inserted in the .txt files provided is rearranged to a tree-search-like structure exemplified in Figure 5, dividing the latitude and longitude values into different categories and eliminating the need to loop through all the elements of the file content to find a searched value of a given latitude and longitude coordinate.

To detect diffraction and reflection points within the first and second Fresnel ellipsoids, two ray-tracing algorithms were designed; in both algorithms, the camera displayed in the Cesium window is used as a starting position, and they can detect both urban building entities and terrain points. The first algorithm, denominated *optical ray-tracing*, only considers the first element intersected in each ray, ignoring the remaining obstacles that may be found after the intersected obstacle. Alternatively, the second algorithm,

denominated *complete ray-tracing*, considers all the obstacles found in a ray direction and inside the Fresnel ellipsoids. To do so, after a ray intersects a building entity, the visible element of each entity is intersected and new rays are cast until either no entities are found or the entities are positioned outside the Fresnel ellipsoids.

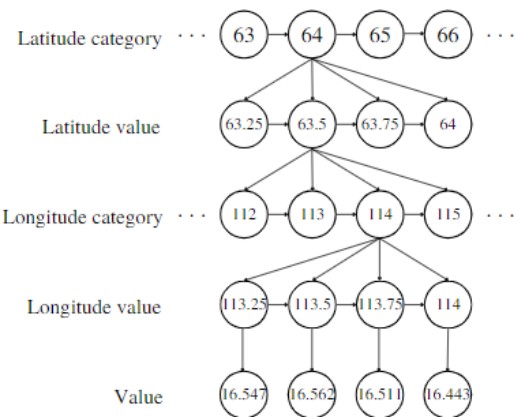

**Figure 5.** Example of created latitude and longitude tree search.

Free-Space Attenuation

To calculate free-space attenuation $L_{bf}$, the ITU-R Recommendation P.525-4 [21] was used. In this recommendation, this attenuation is obtained as follows:

$$L_{bf} = 32.4 + 20 \log(f_{[\text{MHz}]}) + 20 \log(d_{[\text{km}]}) \tag{1}$$

where $f$ is the signal frequency in MHz, and $d$ is the link distance in km.

Diffraction Attenuation

To calculate diffraction attenuation, the ITU-R Recommendation P.526-15 [22] was used. In this recommendation, this attenuation is obtained using three different methods:

- Single knife-edge obstacle, when one obstacle is found;
- Double-isolated edges, when two obstacles are found;
- Bullington model, when three or more obstacles are found.

For **a single knife-edge obstacle**, the diffraction attenuation $J(v)$ is obtained using the following equation:

$$J(v) = -20 \log \left( \frac{\sqrt{[1 - C(v) - S(v)]^2 + [C(v) - S(v)]^2}}{2} \right) \tag{2}$$

where $v$ is defined as

$$v = h \sqrt{\frac{2}{\lambda} \left( \frac{1}{d_1} + \frac{1}{d_2} \right)} \tag{3}$$

where $\lambda$ is the used wavelength, and $h$, $d_1$, and $d_2$ can be obtained following the images provided in P.526-15 [22], and $C(v)$, and $S(v)$ represent the Fresnel integrals, also described in the recommendation.

For **double-isolated edges**, the diffraction attenuation $L$ is obtained using the following equation:

$$L = L_1 + L_2 + L_c \tag{4}$$

or

$$L = L_1 + L_2 - T_c \tag{5}$$

where $L_1$ represents the signal diffraction caused by the first obstacle in an imaginary connection between the transmission antenna and the second obstacle, and $L_2$ represents

the signal diffraction caused by the second obstacle in an imaginary connection between the first obstacle and the reception antenna. $T_c$ and $L_c$ represent the correction term used. Equation (4) is used if $L_1$ and $L_2$ have a value greater than 15 dB and (5), if otherwise.

For **the Bullington model**, the diffraction attenuation $L_b$ is obtained as follows:

$$L_b = L_{uc} + [1 - \exp(-L_{uc}/6)](10 + 0.02d) \tag{6}$$

where $L_{uc}$ represents the knife-edge loss obtained as described in [22].

Atmospheric Attenuation

The model to predict the atmospheric attenuation is described in ITU-R Recommendation P.676-13 [23] and is obtained as follows:

$$A_a = \gamma \times r_0 = 0.1820 \times f \times (N''_{Oxygen}(f) + N''_{WaterVapour}(f)) \times r_0 \tag{7}$$

where $r_0$ represents the path length, $N''_{Oxygen}(f)$ and $N''_{WaterVapour}(f)$ are the imaginary parts of the frequency-dependent complex refractivities.

Rain Attenuation

As defined in ITU-R Recommendation P.838-3 [24], the attenuation due to rain in relation to the rain rate $R$ is obtained as follows:

$$A_r = d \times \gamma_R = d \times kR^\alpha \tag{8}$$

where the coefficients $k$ and $\alpha$, in cases of frequencies between 1 and 1000 GHz, are obtained as follows:

$$\log_{10} k = \sum_{j=1}^{4} \left( a_j \exp\left[ -\left( \frac{\log_{10} f - b_j}{c_j} \right)^2 \right] \right) + m_k \log_{10} f + c_k \tag{9}$$

$$\alpha = \sum_{j=1}^{5} \left( a_j \exp\left[ -\left( \frac{\log_{10} f - b_j}{c_j} \right)^2 \right] \right) + m_\alpha \log_{10} f + c_\alpha \tag{10}$$

The values of the constants $a_j$, $b_j$, $c_j$, $m_k$, $m_\alpha$, $c_k$, and $c_\alpha$ follow Tables 1–4 in Recommendation P.838-3 [24].

Signal Fading

Using the method described in ITU-R Recommendation P.530-18 [25], the percentage of the time during which the enhancement $A$ (dB) is not exceeded, $p_W$ can be obtained as follows:

$$p_W = 100 \left[ 1 - \exp\left( -10^{-q_a A/20} \right) \right] \tag{11}$$

Signal Reflection

Applying the method presented by C. Salema in [26], given the transmission and reception antennas gains $g_E$ and $g_R$, a system frequency $f$ and emission power $p_E$, the dispersed power $dp_s$ for a given point P, which is categorized by Cartesian coordinates depicted in Figure 6, and with an area element of $dxdy$, the signal reflection $dp_s$ can be obtained as follows:

$$dp_s = p_E \times \frac{\lambda^2}{4\pi} \times \frac{g_E^P}{4\pi(x^2 + y^2 + h_E^2)} \times \frac{g_R^P}{4\pi[(d - x)^2 + y^2 + h_R^2]} \times \sigma \times dxdy \tag{12}$$

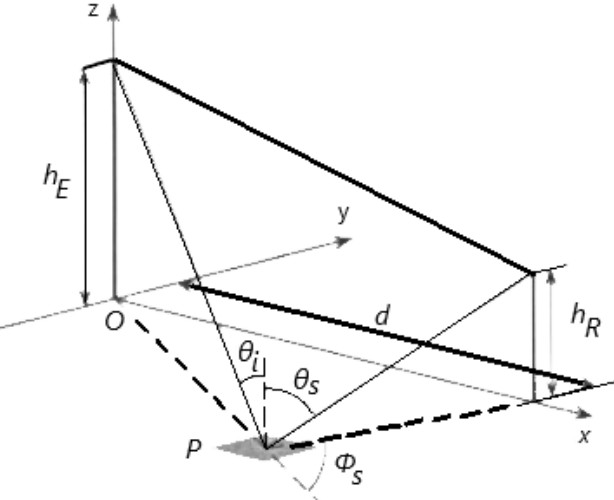

**Figure 6.** Signal reflection coordinates system, as presented by C.Salema in [27].

3.4.5. Link Quality

In this step, the calculated values for the quality metrics SESR, BBER, ESR, and unavailability and respective clauses are presented to the user. Given that the feature to select multiple signal frequencies is presented to the user, the results of these metrics are presented in a two-dimensional chart (Figure 7) using the Plotly.Js library [28], depicting the evolution of the metric values with the difference in signal frequency, as well as in separate output sections for the different metrics (Figure 8).

Using the information presented in the graphic depicted in Figure 7, the optimal frequency is outputted to the user, as it represents the signal frequency with a higher critical margin.

**Figure 7.** Critical margin window (1).

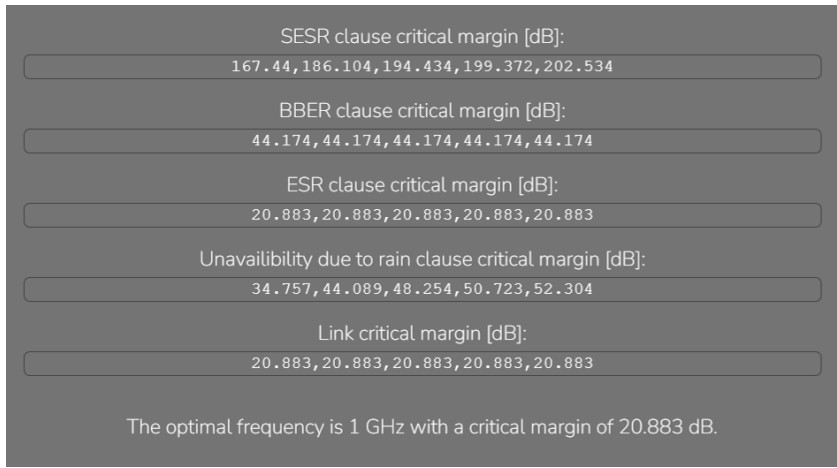

**Figure 8.** Critical margin window (2).

*3.5. Satellite Links*

3.5.1. Antenna Characteristics

In this step, the user is required to input the antenna positions in geographic coordinates. Upon completing the insertion of data, an antenna entity is displayed in a Cesium window provided on the user page.

Using the orbit later selected, the elevation and azimuth angles of each antenna are also outputted in order to be directed to the satellite position.

3.5.2. Orbit Selection

In this step, a set of preconfigured orbits is offered to the user in order to define the satellite orbit that the communication system will use as its base. The information about the current active satellites provided by CelesTrak at http://celestrak.org/NORAD/elements/active.txt (accessed on 27 December 2022) is fetched using the JavaScript fetch API and used to create these orbits. Over 3000 satellite orbit TLE coordinates are provided in this text file, which can be individually displayed in the Cesium window, upon user selection. As the user selects one of the provided orbits, the satellite entity is displayed in the Cesium window, allowing the user to see the possible satellite positions with the evolution of time. Additionally, the user can visualize the complete selected orbit by pressing the provided button, as displayed in Figure 9.

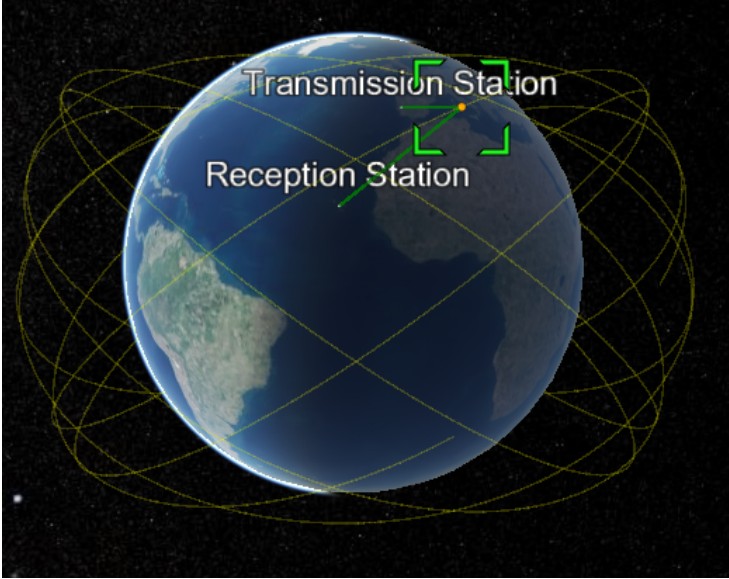

**Figure 9.** Satellite orbit elements in Cesium window.

In Figure 9, two direct lines are also shown, which connect the antenna positions defined in the previous subsection. These direct lines are used as visual queues to indicate if the link has an uninterrupted LoS between the uplink and downlink endpoints; the red color indicates that the LoS is interrupted, and the green indicates otherwise.

This verification was performed using the general formula of intersection between a sphere and a vector. In this case, the vector is composed between the mentioned link endpoints and the satellite position, and the sphere is representative of Earth. The intersections found can result in the following three cases:

- Only one intersection is found, resulting in no signal blockage (case a);
- Two intersections are found, and the antenna position consists of a further intersection, having signal blockage (case b);
- Two intersections are found, and the antenna position consists of the closest intersection, having no signal blockage (case c).

The three cases are depicted in Figure 10.

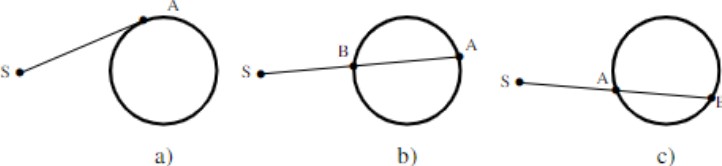

**Figure 10.** Possibilities for signal blockage due to Earth's surface.

### 3.5.3. Satellite Characteristics

In this step, the user is required to input the satellite characteristics required for link budget calculations. The values requested are as follows:

- Satellite power density for transponder saturation;
- Receptor merit factor;
- Saturated effective isotropic radiated power;
- Input back-off with clear sky;
- Output back-off with clear sky.

### 3.5.4. Radio Characteristics

Similarly to terrestrial links, in this step, the user is required to input the radio communication characteristics used in the simulated link. Although maintaining the layout of the terrestrial links, for satellite links, the user can define different frequencies for the ascending link (uplink) and descending link (downlink).

### 3.5.5. QoS Parameters

Here, the user must specify the maximum bit-error ratio (BER) to be reached, as well as the annual maximum time percentage of the outage target. These characteristics have an impact on both the estimates for the link budget as a whole as well as the path loss calculations (particularly the rain attenuation). In order to offer an updated result, these elements are recalculated in response to the changes in this step.

### 3.5.6. Signal Attenuation

For satellite links, only rain and atmospheric attenuations were considered. Although retaining the layout of the terrestrial, this step is divided into uplink and downlink subsections, as the frequencies used to perform the calculation may differ.

#### Atmospheric Attenuation

To calculate rain attenuation, ITU-R Recommendation P.618-13 [29] was used. P.618-13 states that for links using a frequency less than 10 GHz, atmospheric attenuation can be neglected, and for links using higher frequencies, the method used for atmospheric attenuation calculations is the same used in terrestrial links, described in Recommendation P.676 [23].

Rain Attenuation

Recommendation P.618-13 presents a step-by-step method to calculate the rain attenuation in satellite links, where the rain attenuation is obtained as follows:

$$r_{0.01} = \frac{1}{1 + 0.78 \times \sqrt{\frac{L_G \times \gamma_R}{f}} - 0.38 \times (1 - e^{-2 \times L_G})} \tag{13}$$

where $L_G$ represents the horizontal projection of the slant path, $\gamma_R$ is the product found using Equation (8), and $f$ is the radio frequency used.

### 3.5.7. Link Budget

In this section, using all the elements previously inserted for the user and calculations made in the previous sections, the minimal antenna diameters, radiation efficiency, gain, and emission power to assure a reliable connection, both on ascending and descending links, are presented.

The user can also change the value of radiation efficiency to analyze the effects of this characteristic on the remaining elements, using the interface depicted in Figure 11.

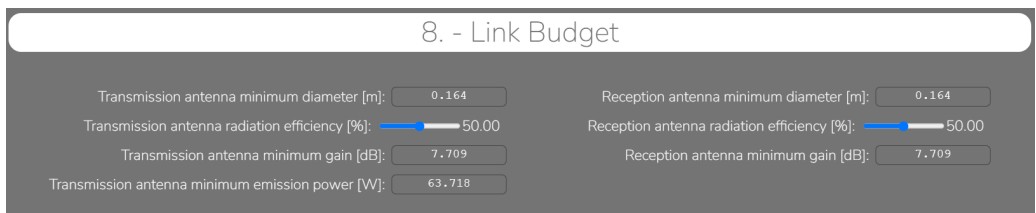

**Figure 11.** Satellite project link budget step window.

## 4. Use Case

In this section, a terrestrial and satellite link project is designed using the software developed. The system parameters are presented, as well as the results outputted using the tool.

### 4.1. Terrestrial Project

This subsection presents the design of a terrestrial RF communication project between a station located in Moscavide (Lisbon, Portugal) and a station located in Palmela (Setúbal, Portugal), using the characteristics shown in Table 2.

**Table 2.** Simulated terrestrial link characteristics.

| | |
|---|---|
| Frequencies (GHz) | 1, 3, 5, 7, 9 |
| Antennas radiation efficiency (%) | 50 |
| Transmission power (W) | 20 |
| Transmission rate (Mbits/s) | 140 |
| Bandwidth (MHz) | 34 |
| Roll of factor | 0.142 |
| Modulation | 64-QAM |
| Polarization | Horizontal |
| Residual BER (RBER) | $1 \times 10^{-12}$ |
| Quality of service (QoS) | X = 0.08 |
| Additional attenuations (dB) | 0 |

Figure 12 displays the transmission and reception antenna positions, as well as the signal route.

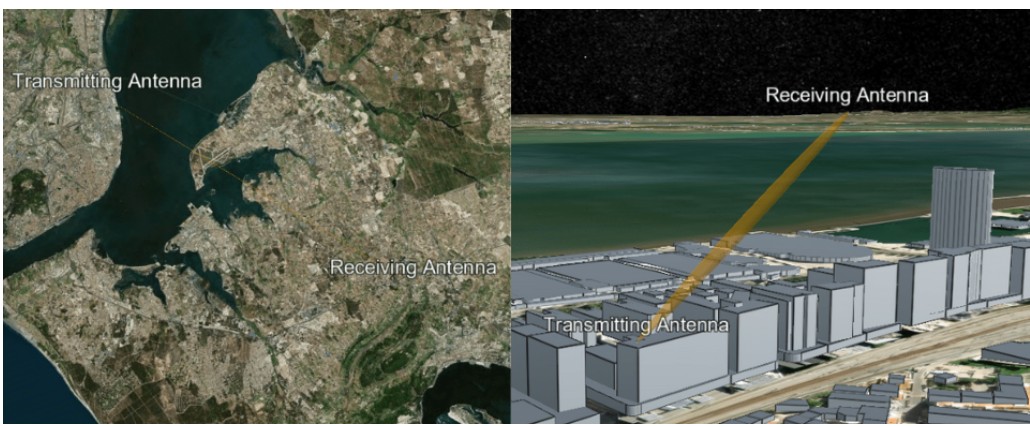

**Figure 12.** Simulated terrestrial project link.

With the given system characteristics and using the modules for signal attenuation mentioned, the results of simulated signal attenuation are presented in Table 3.

**Table 3.** Path attenuations for the simulated terrestrial system.

| Frequency (GHz) | 1 | 3 | 5 | 7 | 9 |
|---|---|---|---|---|---|
| Free-space attenuation (dB) | 121.3 | 120.9 | 135.3 | 138.2 | 140.4 |
| Obstacle attenuation (dB) | 3.4 | 3.4 | 3.4 | 3.4 | 3.4 |
| Atmospheric attenuation (dB) | 0.1 | 0.3 | 0.5 | 0.8 | 1.2 |
| Rain attenuation (dB) | 0.03 | 0.03 | 2.7 | 11.2 | 24.96 |
| Signal reflections (dB) | −65.9 | −17.0 | 12.9 | 29.7 | 38.97 |
| SNR in IPC (dB) | 65.2 | 74.1 | 75.8 | 69.6 | 57.4 |

Analyzing the system's signal reflections, we found that the path contained a high level of signal reflections when frequencies equal to or above 5 GHz were used, resulting in the need to redirect the signal for the system to provide a reliable connection. Therefore, a passive repeater was added, creating a new signal route displayed in Figure 13.

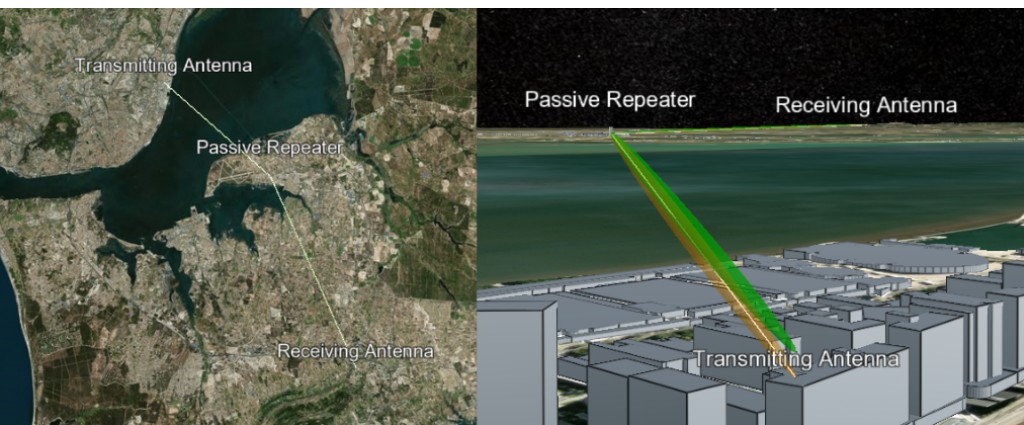

**Figure 13.** Simulated terrestrial project link with passive repeater.

The simulated results with the inclusion of the passive repeater are presented in Table 4.

**Table 4.** System elements with passive repeater addition for the simulated terrestrial project.

| Frequency (GHz) | 1 | 3 | 5 | 7 | 9 |
|---|---|---|---|---|---|
| Free-space attenuation (dB) | 230.7 | 249.8 | 258.7 | 264.5 | 268.9 |
| Obstacles attenuation (dB) | 0 | 0 | 0 | 0 | 0 |
| Atmospheric attenuation (dB) | 0.1 | 0.3 | 0.5 | 0.8 | 1.2671 |
| Rain attenuation (dB) | 0.03 | 0.4 | 2.7 | 11.4 | 25.3 |
| Signal reflections (dB) | $-\infty$ | $-\infty$ | $-\infty$ | $-\infty$ | $-\infty$ |
| SNR in IPC (dB) | 26.8 | 45.3 | 51.3 | 47.98 | 37.8 |

Based on the outputted results, the signal reflection was no longer a degrading element to the system.

To establish a reliable communication system, the system should verify all defined margins, in addition to having a system critical margin above 3 dB [12]. For the simulated project, the verification of project clauses is presented in Table 5.

**Table 5.** Project clause fulfillment for the simulated terrestrial project.

| Frequency (GHz) | 1 | 3 | 5 | 7 | 9 |
|---|---|---|---|---|---|
| SESR clause fulfillment | False | True | True | True | False |
| BBER clause fulfillment | False | False | True | False | False |
| ESR clause fulfillment | True | True | True | True | True |
| Unavailability due to rain clause fulfillment | True | True | True | True | False |

Based on the clause analysis, the critical margins for these clauses were calculated, which are presented in Table 6.

**Table 6.** The critical margins of project clauses for the simulated terrestrial project.

| Frequency (GHz) | 1 | 3 | 5 | 7 | 9 |
|---|---|---|---|---|---|
| SESR critical margin (dB) | −11.96 | 1.7 | 5.6 | 0.7 | −10.6 |
| BBER critical margin (dB) | −16.3 | −2.6 | 1.2 | −3.6 | −14.9 |
| ESR critical margin (dB) | 4.2 | 15.6 | 17.2 | 15.0 | 5.5 |
| Unavailability due to rain critical margin (dB) | 1.3 | 19.5 | 23.7 | 13.6 | −7.4 |
| Project critical margin (dB) | −16.3 | −2.6 | 1.2 | −3.6 | −14.9 |

As presented in both Tables 5 and 6, all clauses were fulfilled only when a frequency of 5 GHz was used. Conversely, using this system frequency did not fulfill a project critical margin of 3 dB (having a 1.2 dB project critical margin), as previously mentioned. Subsequently, diversity and signal equalization was used.

The introduction of diversity and equalization led to the requirement to obtain a 3 dB threshold for the project's critical margin. For this system, both methods of diversity were implemented, as well as signal equalization. These methods were implemented using the characteristics shown in Table 7.

As a result of diversity and equalization implementations, the project's critical margins were recalculated, and the obtained values are presented in Table 8.

Using this method, the project's critical margin with 5 GHz communication frequency was increased to 4.0 dB, thus creating a reliable communication system using the elements and characteristics described throughout the design of the project.

**Table 7.** Diversity and equalization characteristics for the simulated terrestrial project.

| | | |
|---|---|---|
| Space diversity | Distance between space diversity antenna and transmission antenna (m) | 10 |
| | Gain ratio between space diversity antenna and transmission antenna | 1 |
| Frequency diversity | Carriers frequency separation (GHz) | 4 |
| | Gain ratio between frequency diversity antenna and transmission antenna | 1 |
| Signal equalization | Minimum phase gain factor | 490 |
| | Non-minimum phase gain factor | 35 |

**Table 8.** The critical margins of project clauses for the simulated terrestrial project using diversity and equalization.

| Frequency (GHz) | 1 | 3 | 5 | 7 | 9 |
|---|---|---|---|---|---|
| SESR critical margin (dB) | −37.3 | −0.5 | 11.7 | 4.97 | −15.4 |
| BBER critical margin (dB) | −44.98 | −8.2 | 4.0 | −2.7 | −23.1 |
| ESR critical margin (dB) | −23.8 | 12.13 | 18.3 | 15.9 | −1.9 |
| Unavailability due to rain critical margin (dB) | 1.3 | 19.454 | 23.7 | 13.6 | −7.4 |
| Project critical margin (dB) | −44.99 | −8.2 | 4.02 | −2.7 | −23.1 |

*4.2. Satellite Project*

In this subsection, a satellite communication system is simulated using the developed tool. The tool, at the conclusion of the design phase, outputted the necessary antenna characteristics required to ensure a reliable connection.

The simulated satellite system characteristics are presented in Table 9.

**Table 9.** Simulated satellite-link characteristics.

| | |
|---|---|
| Transmission antenna coordinates | 39°23′59″ N–8°1′0″ W |
| Receptive antenna coordinates | 16°0′7″ N–24°0′0″ W |
| Uplink frequencies (GHz) | 2, 4, 6, 8, 10 |
| Downlink frequencies (GHz) | 2, 4, 6, 8, 10 |
| Transmission rate (Mbits/s) | 60 |
| Bandwidth (MHz) | 36 |
| Residual BER (RBER) | $10^{-4}$ |
| Unavailability year percentage (%) | 0.15 |
| Modulation | 4-QAM |
| Uplink polarization | Horizontal |
| Downlink polarization | Horizontal |

For the present simulation, the orbit of satellite LCS 1 was used, with the main Keplerian elements listed in Table 10.

The connection made between the satellite and the stations is depicted in Figure 14.

Following the methods presented and using the weather characteristics outputted, the attenuation values were determined, which are presented in Table 11.

Using all the inputted and outputted elements, in this step, the antenna characteristics were determined to establish a reliable communication system. Considering the

elements presented for this simulation, the antenna characteristics outputted are presented in Table 12.

**Table 10.** Main Keplerian elements of used satellite orbit for the simulated satellite project.

| | |
|---|---|
| Longitude of ascending node (°) | 231,55 |
| Orbit inclination (°) | 32,14 |
| Argument of perigee(°) | 137,36 |
| Ellipse semi-major axis (km) | 9166,05744 |
| Eccentricity | 0,0012574 |

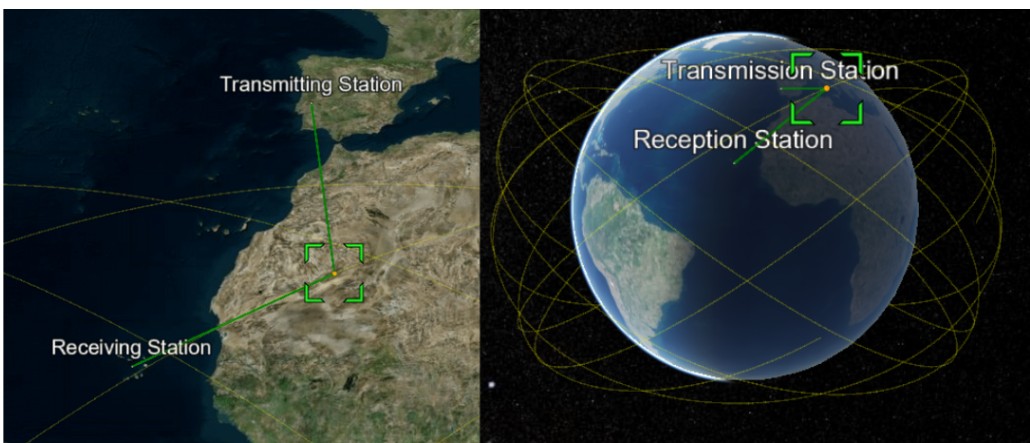

**Figure 14.** Antennas and satellite positions for the simulated satellite project.

**Table 11.** Path attenuations for the simulated satellite system.

| | Frequency (GHz) | 2 | 4 | 6 | 8 | 10 |
|---|---|---|---|---|---|---|
| | Free-space attenuation (dB) | 169.2 | 175.2 | 178.7 | 181.2 | 183.1 |
| Uplink | Atmospheric attenuation (dB) | 0.1 | 0.2 | 0.3 | 0.5 | 0.7 |
| | Rain attenuation (dB) | 0.04 | 0.3 | 1.4 | 3.5 | 6.1 |
| | Free-space attenuation (dB) | 169.9 | 175.9 | 179.4 | 181.9 | 183.9 |
| Downlink | Atmospheric attenuation (dB) | 0.2 | 0.4 | 0.7 | 1.1 | 1.6 |
| | Rain attenuation (dB) | 0.06 | 0.4 | 1.9 | 4.8 | 8.6 |

**Table 12.** Simulated satellite project link budget elements.

| | | |
|---|---|---|
| | Diameter (m) | 1.41 |
| Transmission antenna | Radiation efficiency (%) | 50 |
| | Minimum gain (dB) | 40.4 |
| | Minimum emission power (W) | 61.57 |
| | Diameter (m) | 1.41 |
| Receiving antenna | Radiation efficiency (%) | 50 |
| | Minimum gain (dB) | 40.37 |

## 5. Tool Optimization

In a combined effort with the Portuguese Army Directorate of Communications and Information Systems (DCSI), usability tests were performed in order to evaluate the developed tool's user-friendliness and to define the additional features that were found

useful for using the tool in professional practice. This process granted further professional value to the developed tool through the addition of the suggested features, as well as assuring that the tool is easy to use for the professional user.

During several meetings, the following problems were brought up:

1. The process of adding antennas and passive repeaters in terrestrial projects was found to be difficult since in the first iteration, these elements were added not by using different buttons but by using the right and left click of a mouse; the transmission and receiving antennas were added by order (i.e., first the transmission was added and then the receiving antenna, looping this order to change antenna positions), using the left mouse click, and the passive repeater was added by using the right mouse click. To facilitate this process, the buttons presented at the bottom of Figure 3 were created.

2. Users found it difficult to differentiate an input element from an output element. Therefore, the layout of the different elements was changed; input elements had a white outline, and output elements had a black outline.

3. It was requested to add a numerical element for each step window, allowing the user to understand the project progress made. This feature was added in the title of each step window.

4. The software allowed the user to input the elements without verification. The input of negative values and different decimal separation elements (. or ,) led to inputs that were not valuable and displayed errors to the user, as presented in Figure 15.

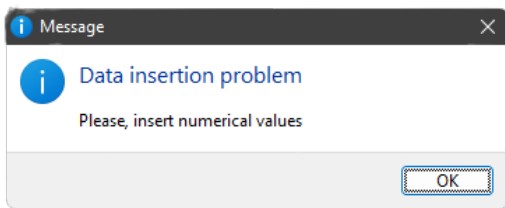

**Figure 15.** Invalid input error message.

5. A step-by-step layout was suggested in order to visualize each step independently of the remaining steps, verifying whether all the inputs of a given step were correctly defined before moving to the next step. This created a methodical process to define the project elements, in contrast to defining different parts of a given step without particular order. To this end, an additional option was added to the project creation window that, when selected, "next" and "previous" buttons would be added to the bottom of the project window, which would change the step presented to the user to the next or previous when clicked, as presented in Figure 16.

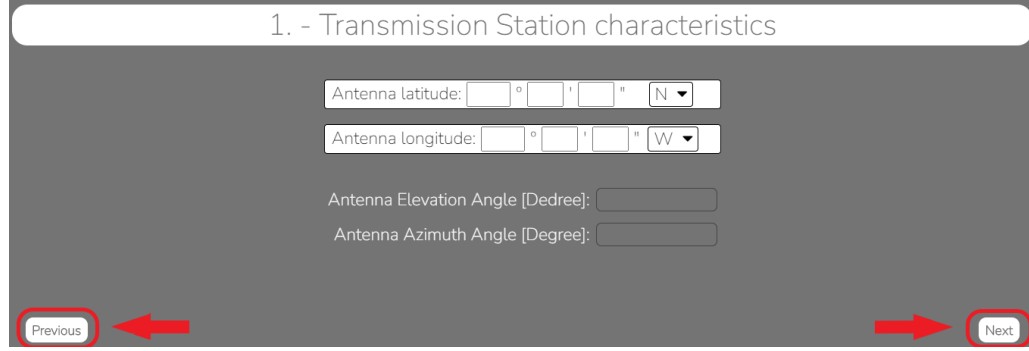

**Figure 16.** Satellite project link budget step window.

Figure 17 presents the project creation window with the mentioned feature highlighted.

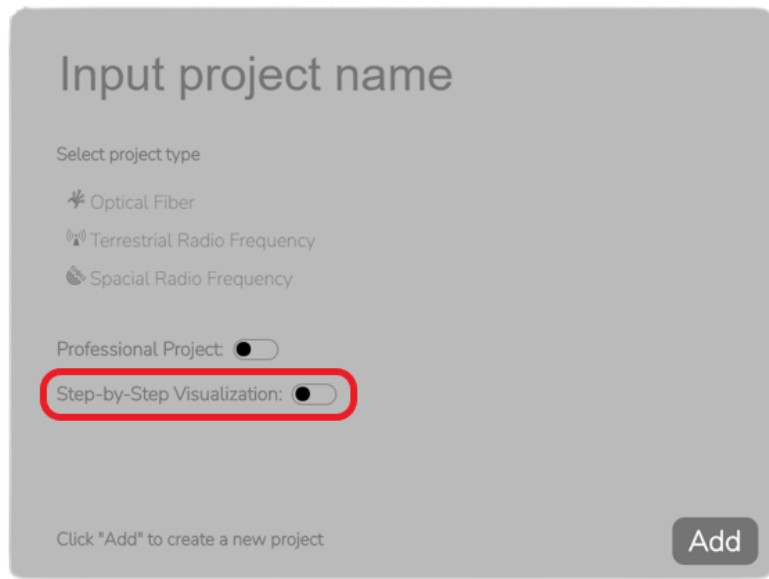

**Figure 17.** Step-by-step project layout option.

6. To facilitate the project design for professional users, it was requested to reduce the number of outputs given by the software. In order to meet this need without removing the academic value of the project design, an additional project layout was created. In this project, the outputs considered unnecessary for professional use were hidden from the user, thus presenting fewer steps to complete the project design.

For terrestrial projects, the following hidden steps were included:

- Equipment reliability;
- Obstacle attenuation;
- Atmospheric attenuation;
- Rain attenuation;
- Unavailability distribution;
- Fast fading;
- Signal reflections;
- SNR in IPC;
- Uniform fading margin;
- Selective fading margin.

For satellite projects, the following hidden steps were included:

- Free-space attenuation;
- Atmospheric attenuation;
- Rain attenuation.

To select this project layout, upon creating a project, the user is presented with the option to create a professional project, as seen in Figure 18.

7. It was requested to create a mechanism that allowed the user to identify in which step it is currently being displayed, in relation to the total steps presented in the project design. To accomplish this goal, a slider was added to the top of the project window, indicating the steps that the current project has and the step currently being presented to the user. The slider also displays which step each element represents by hovering the mouse on each step number and allowing the user to select the said step, scrolling or skipping to the selected step in a normal project or with a step-by step-layout, respectively. The developed feature is presented in Figure 19.

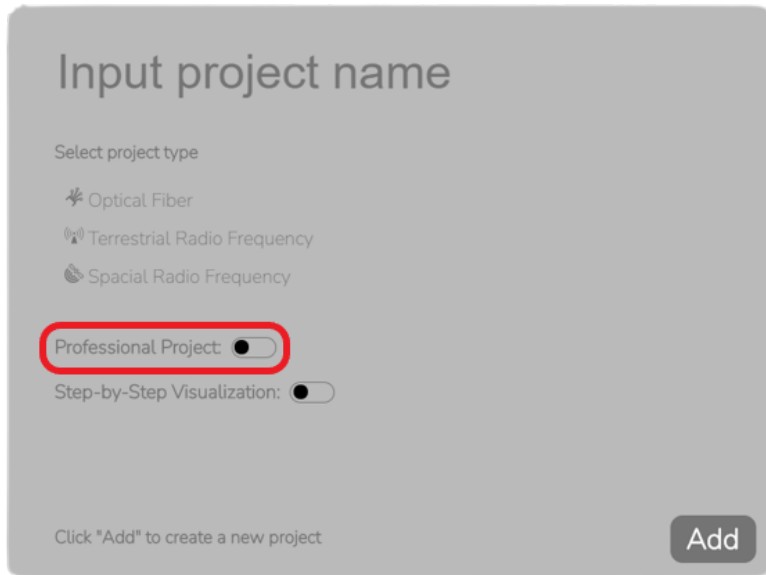

**Figure 18.** Professional project layout option.

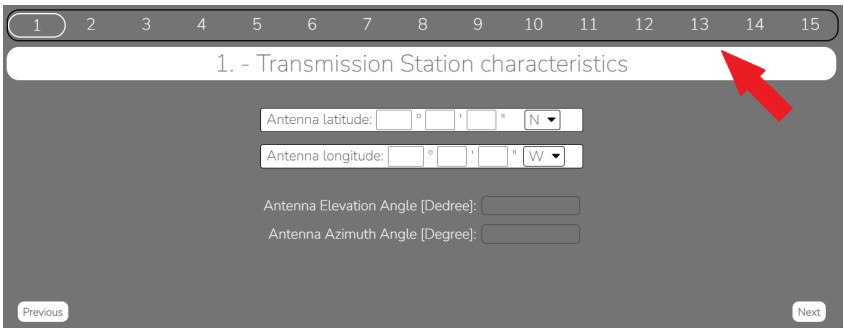

**Figure 19.** Project step display feature.

## 6. Conclusions

In this paper, a tool was developed that is able to automatize the process of the project design of terrestrial and satellite microwave communication systems, thus minimizing the time spent on this phase of the project. This tool abides by the most recent ITU-R Recommendations at the time of development.

Apart from performing the needed calculations, the tool also provides processes to warn the user when an element of the project design is detrimental to the project, as well as how to fix the problem found.

Even though there are tools currently available for similar use, it was found that the majority of the tools available are outdated or bare a large financial investment for engineering students and institutions. Contrarily, the developed tool provides a free-to-use experience, hence mostly advantageous to students. Furthermore, the developed tool can be applied for both academic and professional use by providing a greater understanding of design elements for students and a fast and direct approach for professional project designers.

To validate the outputted results, independent previously designed projects were simulated using the developed tool, to verify whether the results in both instances matched.

There was an opportunity to interact with professional users in this field who tested this tool, and the result was very positive, as their feedback enabled us to optimize the tool, adding additional and useful features.

Thus, a new method for obstacle detection was created. The use of a simplified ray-tracing algorithm allowed us to not only detect obstacles in a link path but also to minimize its attenuation on the signal and the corresponding signal reflections that can

be found in the receiving antenna. Despite the level of complexity of this algorithm and having an elevated execution time when compared with two-dimensional algorithms, the developed algorithm exhibits an increased level of precision, by considering all the elements in a link path and path surroundings, which highlights the greater scientific value of the developed tool.

**Author Contributions:** Software, E.F.; Investigation, E.F.; Writing—original draft, E.F.; Writing—review & editing, P.S., F.C., C.S.C. and A.C. All authors have read and agreed to the published version of the manuscript.

**Funding:** This work was funded by FCT/MCTES through national funds and when applicable co-funded by EU funds under the project UIDB/50008/2020.

**Informed Consent Statement:** Informed consent was obtained from each participant before article submission. consent was obtained from each participant before article submission.

**Data Availability Statement:** The data presented in this study are available on request from the corresponding author. The data are not publicly available due to privacy.

**Acknowledgments:** A special thanks to the Portuguese Army Directorate of Communications and Information Systems for welcoming us with open doors and for the combined effort to make the presented tool better in every possible way.

**Conflicts of Interest:** The authors declare that they have no conflict of interest, financial or otherwise.

## Abbreviations

The following abbreviations are used in this manuscript:

| | |
|---|---|
| API | Application programming interface |
| BBER | Background block error ratio |
| BER | Bit-error ratio |
| CSS | Cascading style sheets |
| DCSI | Portuguese Army Directorate of Communications and Information Systems |
| ESR | Errored second ratio |
| GLB | GL transmission format binary |
| glTF | GL transmission format |
| HTML | HyperText markup language |
| IPC | Ideal propagation conditions |
| ITU-R | International Telecommunication Union Radiocommunication Sector |
| JSON | JavaScript object notation |
| LoS | Line of sight |
| NPM | Node Package Manager |
| PSK | Phase-shift keying |
| QAM | Quadrature amplitude modulation |
| QoS | Quality of service |
| RBER | Residual BER |
| SNR | Signal-to-noise ratio |
| TLE | Two-line element set |

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
