# Peer review of "An Optimized Planning Tool for Microwave Terrestrial and Satellite Link Design"

_futureinternet, doi:10.3390/fi15020058_

Round 1

Reviewer 1 Report

Major comments

1.      This manuscript claimed that the authors developed a system that can be used for terrestrial and Spacial Microwave link design. Here, I find the meaning of the word Spacial is not clear for radio wave link design. Even if we assume its purpose pertaining to or involving or having the nature of space, in the literature, people do not use this vocabulary to mean satellite link.

2.      The introduction of this manuscript lacks adequate information to introduce its reader to the topic. In this section, sufficient references are not cited.

3.      In this study, the authors claimed that they had designed a system that could be used to prepare the link budget of terrestrial and spacial point-to-point links. But in the manuscript's contents, the detailed analysis of how the system can be used for a radio link budget design is unclear.

4.      In the manuscript, there is no precise mathematics of how that developed system could be used for designing a radio link budget.

5.      I think this part is completely missing in this manuscript, and it is mandatory to add clear mathematics for publication in a journal.

6.      Most of the references in this manuscript are from ITU guidelines, and very few references are used from the research article. It is recommended that the manuscript should have to cite the up-to-date reference.

7.      The authors have claimed that their system can handle different types of terrestrial and spacial microwave links. In this regard,  they have mentioned —obstacles that can appear either in terrestrial or spacial links, for example atmospheric attenuation or rain attenuation, and many more. But the manuscript lacks any additional analysis about atmospheric or rain attenuation.

8.      I think the author can improve their manuscript by comparing their system with some already implemented systems, and it needs to include how their system is different from other systems.

Minor comments

1.      If the word figure refers to an inserted figure in the manuscript, it needs to use Figure instead of Figure. (lines 201, 203)

2.       Spacing before a sentence in a paragraph. (lines 8, 11)

Reviewer 2 Report

Although the article is in general well written, I have some minor comments prior publication:

1) Please pay special attention to some grammar and syntax errors. For example, in the abstract I read that "There are many people that 1
lives in remote areas"

2) Some figures with no particular added value, such as Figure 2 and Figure 3 can be removed.

3) There is no information on what specific channel models are used in the tool.

4) Following the previous comment, what types of modulation orders can be supported?

5) It is important to provide at least one use case with all calculations for a particular coverage scenario.

6) References list is quite limited and out of date. The following publication can be useful, since it refers to the design and implementation of simulation tools for 5G networks: Gkonis, P.K.; Trakadas, P.T.; Kaklamani, D.I. A Comprehensive Study on Simulation Techniques for 5G Networks: State of the Art Results, Analysis, and Future Challenges. Electronics 2020, 9, 468. https://doi.org/10.3390/electronics9030468

Round 2

Reviewer 1 Report

1: Why are the authors not using "Satellite Link" (which is a well-known term) instead of "Spacial Link" in the manuscript title>

2: The meaning of the sentence in lines 84—87 is not clear. 

3: This sentence is not completed (lines 95—97)—"For the professionally developed tools, the free version was used for TAP 7 (having a yearly license of US$1999) and MLinkPlanner 2.0 (having a yearly license of US$399)". In addition, it is a statement containing information that needs a reference.

4: The literature review should be integrated with the introduction section (section 1).

5: What is the meaning of the question mark (??) in line 140?

6: What is the meaning of "depicted in Figure 3 s" in line 140? 

7: Why for a single sentence written as a new paragraph (line 369)?

8: Many sections start or contain a list (as in line 192), which deserves more explanation.

9: Many references are included without detailed attributes. Therefore it is hard to follow, for example, ref. [8], [11], [12], and more. 
